# Antimicrobial, Anti-Biofilm, Anti-Quorum Sensing and Cytotoxic Activities of *Thymbra spicata* L. subsp. *spicata* Essential Oils

**DOI:** 10.3390/antibiotics14020181

**Published:** 2025-02-11

**Authors:** Timur Hakan Barak, Mujde Eryilmaz, Basar Karaca, Huseyin Servi, Simge Kara Ertekin, Muhittin Dinc, Hatice Ustuner

**Affiliations:** 1Department of Pharmacognosy, Faculty of Pharmacy, Acibadem Mehmet Ali Aydinlar University, Istanbul 34752, Türkiye; timur.barak@acibadem.edu.tr; 2Department of Pharmaceutical Microbiology, Faculty of Pharmacy, Acibadem Mehmet Ali Aydinlar University, Istanbul 34752, Türkiye; 3Department of Biology, Faculty of Science, Ankara University, Ankara 06100, Türkiye; karaca@ankara.edu.tr; 4Department of Pharmacognosy, Faculty of Pharmacy, Istanbul Yeni Yüzyıl University, Istanbul 34010, Türkiye; huseyin.servi@yeniyuzyil.edu.tr; 5Department of Pharmaceutical Toxicology, Faculty of Pharmacy, Istanbul Yeni Yüzyıl University, Istanbul 34010, Türkiye; simge.karaertekin@yeniyuzyil.edu.tr; 6Department of Mathematic and Science Education, Ahmet Keleşoğlu Faculty of Education, Necmettin Erbakan University, Konya 42090, Türkiye; mdinc@erbakan.edu.tr; 7Department of Biology, Faculty of Science, Akdeniz University, Antalya 07058, Türkiye; h.ustuner1977@gmail.com

**Keywords:** *Thymbra spicata* subsp. *spicata*, quorum sensing inhibition, biofilm inhibition, cytotoxicity, essential oil

## Abstract

**Background/Objectives:** Essential oils of *Thymbra spicata* subsp. *spicata* are known for their rich phytochemical content and bioactive properties. This study aimed to evaluate the antimicrobial, anti-biofilm and anti-quorum sensing, as well as the cytotoxic activities of *T. spicata* subsp. *spicata* essential oils (TS-EO1 and TS-EO2) obtained from two different localities in Türkiye, along with a detailed chemical composition analysis. **Methods:** TS-EO1 and TS-EO2 were obtained by the hydrodistillation method and analyzed using Gas Chromatography—Mass Spectrometry (GC-MS) to determine their phytochemical profiles. Antimicrobial activities were assessed against Gram-positive and Gram-negative bacteria, and fungal strains were assessed using the broth microdilution method. Anti-biofilm and anti-quorum sensing activities were evaluated using *Pseudomonas aeruginosa* PAO1 and *Chromobacterium violaceum* ATCC 12472, respectively. Cytotoxic properties were tested on four cell lines (A549, MCF-7, U87MG, and L929) using the MTT assay. **Results:** Both essential oil samples were rich in carvacrol (54.3% and 54.1%), followed by p-cymene and γ-terpinene. The essential oils exhibited significant antimicrobial activity, particularly against *Staphylococcus aureus* (6.25 mg/mL) and *Candida parapsilosis* (0.20 mg/mL). Sub-MIC concentrations significantly inhibited biofilm formation and quorum sensing. Both samples showed moderate cytotoxic properties against human cancer cell lines, particularly A549 (IC50: 116.3 and 134.4 μg/mL, respectively). **Conclusions:** This study showed that *T. spicata* subsp. *spicata* essential oils have significant antimicrobial, anti-biofilm, and anti-quorum sensing properties against various bacteria and fungi, along with moderate cytotoxic effects, indicating their medicinal and pharmaceutical potentials. This is the first study which revealed anti-biofilm and anti-quorum sensing properties of *T. spicata* essential oils to our knowledge.

## 1. Introduction

Widely distributed across the Mediterranean basin, the *Lamiaceae* family comprises a range of herbaceous plants and shrubs which are well regarded in both folk cuisine and medicine. In Türkiye, thymol and carvacrol-rich plants are collectively known as “Kekik” due to their distinctive thyme-like aroma. This group includes various species from the genera Origanum, *Satureja*, *Thymbra*, *Thymus*, and *Coridothymus*, all part of the *Lamiaceae* family [1]. Harvested from the wild, these plants are traditionally used in Turkish cuisine and medicine or exported after drying. Among these, *Thymbra spicata* L., a small shrub of the Eastern Mediterranean region, stands out with high carvacrol content in its essential oils (EOs), lending it potent antimicrobial, antioxidant, and other medicinal properties. Known locally as “Zahter” or “Karabaş kekik”, *T. spicata* grows primarily in Southeastern Anatolia, Thrace, and coastal areas of Türkiye’s Aegean and Mediterranean regions [2]. Ethnobotanical research highlights the widespread use of *T. spicata* across Türkiye for various health concerns. Locally brewed into tea, its aerial parts are traditionally employed to treat ailments such as stomach aches, kidney stones, respiratory infections, gastrointestinal disorders, and cardiovascular issues [3]. The therapeutic applications extend to managing diabetes, hypercholesterolemia, headaches, and inflammatory conditions. In addition to its culinary use, the essential oil-rich plant serves as a remedy for bronchitis, rheumatism, and colic, with carvacrol levels in some plants reaching over 90% of the total oil content [4,5,6].

Recent studies underscore the diverse pharmacological potential of *T. spicata*, including anti-inflammatory, antioxidant, and antimicrobial activities. These benefits are attributed to its rich composition of phenolic compounds, such as rosmarinic acid, carvacrol, and thymol, which enhance its medicinal value [7]. Notably, its essential oils exhibit effective antimicrobial properties, with low minimum inhibitory concentration (MIC) values against common pathogens like *Staphylococcus aureus*, *Bacillus subtilis*, *Enterococcus faecalis*, and *Escherichia coli* [8]. In addition, previous studies demonstrated the cytotoxic potential of various *Thymbra* species taxa, including *T. spicata* subsp. spicata and its major ingredient carvacrol [9].

Despite these findings, there is a lack of comprehensive studies investigating the anti-biofilm and anti-quorum sensing (anti-QS) properties of *T. spicata* essential oils, which are critical for combating biofilm-associated infections and antibiotic resistance. Biofilm formation is a significant challenge in clinical settings, as it provides a protective niche for pathogens, enhancing their survival and resistance to conventional antibiotics [10]. Similarly, QS, a bacterial communication system that regulates biofilm formation and virulence represents an attractive target for novel antimicrobial strategies [11].

In light of the information in this study, two different essential oil samples of *T. spicata* subsp. *spicata* (TS-EO1 and TS-EO2) from the Antalya province were investigated for their antimicrobial and cytotoxic properties. For better evaluation of antimicrobial potential, anti-biofilm and anti-quorum sensing activity were investigated. For elucidating the cytotoxic potential of the samples, IC_50_ values of the samples were detected against four different cell lines. In addition, phytochemical profiles of the samples were revealed with detailed Gas Chromatography (GC) analysis. To our knowledge, this is the first study which revealed anti-biofilm and anti-QS properties of *T. spicata* essential oils.

## 2. Results and Discussion

### 2.1. Phytochemical Analysis of EOs

Phytochemical analysis was done for the EOs of both locations, and a total 23 compounds were detected; in both samples, 18 compounds in the TS-EO1 and 16 compounds at the TS-EO2 and 100% of all ingredients were revealed (Table 1 and Figure 1). See Appendix A.

Results showed that carvacrol is the dominant ingredient for EOs of both samples with very similar amounts of 54.3 and 54.1%, respectively. Even though γ-terpinene was significantly lower than carvacrol, it was determined as the second abundant ingredient for both essential oils, 12.4 and 13.3%, respectively. p-cymene was also determined in significant amounts, 10.1 and 12.3%. Although the phytochemical profiles of the samples are greatly analogous, there are variations between the samples. β-pinene is determined as 3.3% of the TS-EO1, while it was seen at 0.8% in the TS-EO2. Moreover, δ-3-carene, β-phellandrene and β-cis-ocimene were detected only at TS-EO1, Trans-sabinene hydrate, p-tert-butylcatechol and pimara-7,15-dien-3-ol were seen in the TS-EO2. Even if both samples were obtained from the same species and their geographical locations are fairly adjacent, various of other factors can affect the phytochemical ingredients of plants [12]. Various studies were conducted to elucidate profiles of EOs from *T. spicata* previously. Kızıl (2010) investigated variation of *T. spicata* essential oils which are collected from 26 different localities from Türkiye [6]. Results revealed that carvacrol was the major ingredient of 25 of them varying from 34.5 to 94.22%. In the sample from Kulp province, γ-terpinene was detected as the major ingredient with 52.98% while in three of the samples it was detected lower than 4%. In a study conducted by Özdemir Nath et al. (2023) significantly higher amount of carvacrol was detected with 71.05%, and parallel to our results, γ-terpinene and p-cymene was detected as the second and third most abundant ingredients [9].

Stefanaki et al. (2018) observed similar variation in the *T. spicata* essential oils from Greece [13], while Momeni et al. (2020) revealed that soil conditions have a significant effect on the phytochemical profile [14]. Other minor ingredients were also detected in the samples; p-tert-butylcatechol is detected in TS-EO2 in 0.5%, and this is the first detection in *T. spicata*, to our knowledge. However, it was observed as one of the major ingredients in other *Thymbra* species, *T. sintenisii* in a previous study [15]. Trans-Sabinene hydrate is also detected in TS-EO2 as 0.4%, and spathulenol as 0.1% in both samples, which are parallel with the previous reports [7,16,17]. Pimara-7,15-dien-3-ol which is a diterpene, was detected in TS-EO2 which is the first time in *Thymbra* genus to our knowledge, however, it was previously detected in an essential oil from *Lamiaceae* family [18]. Results of the phytochemical evaluation stated that variations and the phytochemical profile of *T. spicata* EOs were in correlation with the existing literature.

### 2.2. Evaluation of the Cytotoxicity Potential

Essential oils are known for their antioxidant and antimicrobial properties; however, their cytotoxic potentials have been shown in previous studies [19]. For this reason, essential oils obtained from *T. spicata* from two different localities were investigated for determination of their cytotoxic properties. It was determined that both samples have complex phytochemical profiles, of which any of the ingredients may contribute to cytotoxic potential. Four different cell lines were used in order to maintain higher accuracy of cytotoxic bioactivity. Results showed that both samples have moderate cytotoxic properties against all four cell lines (Table 2 and Figure 2).

In a previous study, EO from *T. spicata* subsp. *spicata* which carvacrol was the dominant ingredient, showed cytotoxic properties against MCF-7 cell line [20]. Similarly, in another study, the cytotoxic potential of the *T. spicata* subsp. *spicata* was revealed [9]. When considering carvacrol as the predominant ingredient of the samples, it can be hypothesized that it may initiate the cytotoxic properties. In a previous study, it was shown that carvacrol induced apoptosis through p53-dependent and Bcl-2/Bax pathways [21]. Various studies in the literature demonstrated antitumor activity of carvacrol for various cell lines and it was shown that carvacrol had more potent antitumor potential of its isomer, thymol [22]. Synergistic interactions should be considered for better understanding of the bioactivities of EOs thus, they are complex mixtures of various ingredients. Other significant ingredients of the samples p-cymene, γ-terpinene and α-pinene showed significant anticarcinogenic properties against melanoma cell lines [23]. Moreover, it was shown that from several other *Thymbra* species, there was a positive correlation between carvacrol and thymol content and cytotoxic antiproliferative bioactivity [24].

### 2.3. Evaluation of Antimicrobial Activity

The MIC values (mg/mL) of TS-EO1 and TS-EO2 are presented in Table 3 and Table 4. The EOs demonstrated stronger antibacterial activity against Gram-positive bacteria compared to Gram-negative bacteria. Both EOs exhibited the lowest MIC values against *S. aureus* ATCC 29213 (MSSA) and *S. aureus* ATCC 43300 (MRSA), with MIC values of 12.5 mg/mL and 6.25 mg/mL, respectively. These results suggest that TS-EO1 and TS-EO2 possess the strongest antibacterial activity against *S. aureus* strains (Table 3).

Among the tested fungi, the EOs demonstrated the strongest antifungal activity against *C. parapsilosis* RSKK 994, with MIC values of 0.20 mg/mL for TS-EO1 and 0.39 mg/mL for TS-EO2. Overall, TS-EO2 demonstrated stronger antifungal activity compared to TS-EO1 (Table 4).

For the *P. aeruginosa* PAO1 strain, the MIC values for the TS-EO1 and TS-EO2 were both determined to be 25 mg/mL. Sub-MIC concentrations (below 25 mg/mL) of TS-EO1 and TS-EO2 were evaluated for their antibiofilm activity. The results of the antibiofilm activity of the EOs are presented in Figure 3. TS-EO2 demonstrated the strongest antibiofilm activity (Figure 3b). Sub-MIC concentrations of both TS-EO1 and TS-EO2 significantly inhibited biofilm formation (Figure 3a,b). At a concentration of 12.5 mg/mL (MIC/2), TS-EO2 reduced biofilm formation by 88.13% (Figure 3b). The ability of TS-EO1 to inhibit biofilm formation at concentrations of 3.12, 6.25 and 12.5 mg/mL was found to be statistically significant, while this result was observed for TS-EO2 at all concentrations in the range of 0.20–12.5 mg/mL (Figure 3a,b).

Figure 4 illustrates the effects of TS-EO1 and TS-EO2 on the inhibition of violacein at different concentrations for determination of anti-QS activity. A clear dose-dependent increase in inhibition was observed for both EOs. In particular, TS-EO1 showed an increase in inhibitory activity from 38.66% at 0.39 mg/mL to 86.39% at 1.56 mg/mL. Statistical analysis revealed significant differences in the mean inhibitory values at the concentrations tested. Similarly, TS-EO2 showed a dose-response trend; however, no statistically significant differences were observed between the 0.39 mg/mL and 0.78 mg/mL concentrations. Based on the MIC values determined, TS-EO2 showed a stronger effect in reducing violacein production.

TS-EO1 and TS-EO2, which are mainly composed of p-cymene, γ-terpinene and carvacrol, showed significant biological activities, including antimicrobial, antibiofilm and anti-QS effects, which can be attributed to these major constituents. The antimicrobial activity tests showed that the EOs of *T. spicata* (TS-EO1 and TS-EO2) exhibited stronger antibacterial activity against Gram-positive bacteria than against Gram-negative bacteria, with particularly significant inhibition observed against *S. aureus* strains (both MSSA and MRSA). Carvacrol, a well-known phenolic compound found in high concentrations within essential oils (54.1–54.3%), has been previously reported for its potent antibacterial activity. The mechanism of action of carvacrol is based on its ability to disrupt the integrity of the bacterial membrane. As a lipophilic compound, carvacrol is incorporated into the lipid bilayer of the bacterial cell membrane, destabilizing it and increasing membrane permeability. This leads to the leakage of vital cell contents, disrupts cell respiration and ultimately leads to the death of the bacterial cells. The antibacterial effect of carvacrol is particularly strong due to this membrane-directed action, which has been observed in both Gram-positive and Gram-negative bacteria. Studies have shown that carvacrol is particularly effective against *S. aureus*, which aligns with the low MIC values (6.25–12.5 mg/mL) observed in our study [25,26]. The antifungal activity of the EOs of *T. spicata* was more pronounced against *C. parapsilosis* (MIC: 0.20–0.39 mg/mL), indicating a possible use in the control of fungal infections. The antifungal properties of carvacrol and γ-terpinene are well documented, especially their ability to destroy the cell wall of fungi and inhibit the germination of spores. Carvacrol, a major component of many essential oils, acts by interfering with the synthesis of chitin in the fungal cell wall, which is crucial for fungal survival. This disruption leads to an impairment of cell integrity and ultimately to cell death. In addition, carvacrol can inhibit the synthesis of glycoproteins and mitochondrial structures and thus further damage the fungal cells. Similarly, γ-terpinene, another important component of EOs, enhances the antifungal effect by destabilizing the cell membrane and disrupting fungal communication pathways, such as the sporulation and germination processes [27,28,29,30]. Biofilm formation of *P. aeruginosa* is a crucial factor for its pathogenicity and resistance to antibiotics. Our study showed that sub-MIC concentrations of both EOs significantly inhibited biofilm formation of *P. aeruginosa* PAO1. In particular, TS-EO2 reduced biofilm formation by 88.13% at 12.5 mg/mL. Studies have shown that carvacrol at sub-inhibitory concentrations effectively reduces biofilm formation by disrupting the structural integrity of biofilms. Carvacrol interferes with bacterial adhesion and biofilm development through mechanisms such as reducing the production of quorum-sensing molecules and the extracellular matrix. This activity makes carvacrol a promising natural agent to combat biofilm-associated infections, which are usually resistant to conventional treatments [30,31]. In a similar study by Onder et al. (2024), the essential oil of *Mentha longifolia*, which is also rich in carvacrol, showed comparable antibiofilm activity against *Staphylococcus epidermidis* [32]. Quorum sensing (QS) is a bacterial communication mechanism that regulates the production of virulence factors and the formation of biofilms. In this study, both essential oils significantly inhibited the production of violacein in *C. violaceum* ATCC 12472, with TS-EO1 showing 86.39% inhibition at 1.56 mg/mL. Carvacrol and γ-terpinene have been shown to interfere with QS-regulated gene expression in various bacteria, reducing virulence and biofilm formation. Carvacrol, for example, inhibits the production of QS molecules, which are essential for bacterial communication and biofilm formation. Studies have shown that sublethal concentrations of carvacrol reduce bacterial motility and QS-regulated biofilm formation, while γ-terpinene has similar QS-inhibitory effects and further reduces bacterial virulence [31,32,33].

While the MIC values of *T. spicata* EOs appear high compared to standard antibiotics and antifungals, this observation is consistent with previous studies on plant-derived natural products. EOs are complex mixtures of bioactive compounds that function through multi-target mechanisms, often requiring higher concentrations to achieve effective microbial inhibition. Unlike synthetic antibiotics, which are designed to act on specific targets, EOs disrupt multiple cellular processes, including membrane integrity, protein synthesis, and QS regulation. Additionally, the hydrophobic nature of essential oils influences their solubility and bioavailability in in vitro conditions, which may contribute to the higher MIC values observed. Despite this, essential oils offer advantages such as lower potential for resistance development and synergistic effects with conventional antimicrobials, making them promising candidates for future therapeutic applications [27].

The findings of this study highlight the significant antimicrobial, anti-biofilm and anti-QS properties of *T. spicata* EOs, underscoring their potential for application in clinical and pharmaceutical settings. Given the increasing global concern regarding antibiotic-resistant infections, the observed inhibition of QS suggests that *T. spicata* EOs could serve as alternative agents for reducing bacterial virulence.

## 3. Materials and Methods

### 3.1. Plant Materials and Obtaining EOs

The specimens of *T. spicata* subsp. *spicata* (zahter) were collected from two different localities in Antalya province. The localities; Antalya: Manavgat, Amedler Köyü road, *Pinus brutia* clearings, 250–300 m, 28.07.2023, M. Dinç 3666 & H. Üstüner. Antalya: Beşkonak, Köprülü Kanyon, *Pinus brutia* clearings, 50 m, 28.07.2023, M. Dinç 3668 & H. Üstüner. The specimens were dried according to the standard herbarium procedures, identified by Prof. Dr. Muhittin Dinç and deposited in the Selçuk University Faculty of Science Herbarium (KNYA). The plant materials were dried at appropriate conditions in the laboratory. The dry aerial parts *T. spicata* L. subsp. *spicata* were crumbled into small pieces and soaked with distilled water (1000 mL), then extracted by hydrodistillation for 3 h, using a Clevenger apparatus.

### 3.2. Phytochemical Analysis of EOs

The EOs were analyzed by GC-MS (Agilent 5975C Inert XL El/Cl MSD System, Agilent Technologies, Santa Clara, CA, USA) and GC-FID using polar and non-polar columns: HP-5MS (5% phenyl, 95% methyl polysiloxane; 30 m × 0.25 mm, 0.25 m film thickness). The oven temperature was programmed as follows: isothermal at 60 °C for 1 min, then increased to 246 °C, at a rate of 3 °C min^−1^ and subsequently held isothermal for 30 min. The carrier gas was helium, with a flow rate of 0.9 mL min^−1^ [34]. The relative percentages of the separated compounds were calculated from the integration of the peaks shown in FID chromatograms and the identification of the compounds was done by relative retention indices comparison of n-alkane series to the literature and with mass spectra comparison with GC libraries and some of the samples also identified through reference standards [35] (Wiley 8th Ed./NIST05 and NIST14 Mass Spectra libraries). Reference compounds were purchased from Sigma-Aldrich (St. Louis, MO, USA).

### 3.3. Evaluation of Cytotoxicity of EOs

The cell lines used to determine the anticancer effect were A549 (human lung adenocarcinoma), MCF-7 (human breast carcinoma), U87 (human glioblastoma), and L929 (mouse fibroblast cell), all obtained from ATCC. The cells were cultured in DMEM supplemented with 10% fetal bovine serum and 1% penicillin-streptomycin and maintained at 37 °C in a humidified atmosphere containing 5% CO_2_. Two essential oils were prepared by dissolving them in ethanol at a concentration of 2.5 mg/100 µL, ensuring the final ethanol concentration was below 1%. For the MTT assay, cells were seeded into 96-well plates at a density of 5 × 10⁴ cells per well and treated with varying concentrations of the essential oils (10–250 µg/mL) for 48 h. Following the treatment period, 10 µL of MTT solution (5 mg/mL in PBS) was added to each well, and the plates were incubated in the dark for 4 h. The resulting formazan crystals were dissolved in DMSO, and absorbance was measured at 570 nm using a microplate reader to calculate cell viability [36].

### 3.4. Antimicrobial Activity Test

In the antimicrobial activity assays, the bacterial strains *Staphylococcus aureus* ATCC 29213 (methicillin-susceptible, MSSA), *S. aureus* ATCC 43300 (methicillin-resistant, MRSA), *Enterococcus faecalis* ATCC 29212, *Escherichia coli* ATCC 25922, and *Pseudomonas aeruginosa* ATCC 27853 were used. Fungal strains included *Candida parapsilosis* RSKK 994, *C. parapsilosis* ATCC 22019, *C. glabrata* RSKK 4019, *C. krusei* RSKK 3016, and *C. albicans* ATCC 10231. The broth microdilution method was performed to determine the Minimum Inhibitory Concentration (MIC) values of the *T. spicata* EOs [37,38].

In accordance with the recommendations of Onder et al. (2024), essential oil samples were prepared for antibacterial and antifungal activity tests [32]. For the antibacterial activity test, TS-EO1 and TS-EO2 were serially diluted 2-fold to obtain concentrations ranging from 50 mg/mL to 0.78 mg/mL in Mueller-Hinton Broth (MHB; Difco Laboratories, Detroit, MI, USA), which was supplemented with 0.5% (*v*/*v*) Tween 80 (Merck, Darmstadt, Germany) to facilitate dissolution of the samples. The wells containing only inoculum and MHB with Tween 80 served as the control group. The inoculums were prepared from overnight cultures and the final concentrations of the test bacteria were adjusted to 5 × 10^5^ CFU/mL in each well. The microtiter plates were then incubated at 35 °C for 18–24 h. The MIC values were determined as the lowest concentration (mg/mL) of EOs at which no visible bacterial growth was observed. Ciprofloxacin (Sigma, St. Louis, MO, USA) was used as the reference antibiotic for comparison.

For the antifungal activity tests, the EOs were dissolved in a solution of 10% dimethyl sulfoxide (DMSO) with 0.02% Tween 80. The concentration range was then prepared in RPMI 1640 broth (ICN-Flow, Aurora, OH, USA) supplemented with glutamine but without bicarbonate and pH indicator as previously described. The control wells contained the modified culture medium and inoculum. The fungal inoculum was adjusted to a final concentration of 0.5–2.5 × 10^3^ CFU/mL in each well. The plates were incubated at 35 °C for 48 h. The MIC values were determined as the lowest concentration (mg/mL) of EOs at which no visible fungal growth was observed. Amphotericin B (Sigma, USA) was used as the reference standard for comparison.

### 3.5. Antibiofilm Activity Test

The MIC values of both EOs for *P. aeruginosa* PAO1 were determined before the antibiofilm activity test. The MIC/2 values of each essential oil were considered for the dilution process and these MIC/2 concentrations were diluted in Brain Heart Infusion Broth (BHI; Merck, Darmstadt, Germany) supplemented with 2% sucrose and 0.5% Tween 80. A modified crystal violet assay was used to confirm the antibiofilm activity of the EOs against *P. aeruginosa* PAO1 in vitro using a microplate-based biofilm model [39,40].

*P. aeruginosa* PAO1 was initially cultured in 5 mL of Brain Heart Infusion (BHI) broth and incubated at 37 °C for 24 h. Following incubation, the bacterial suspension was adjusted to a concentration of approximately 10^6^ CFU/mL in BHI supplemented with 2% sucrose. A 10 µL aliquot of this inoculum was added to each well, along with 140 µL of modified medium containing sub-MIC concentrations of the EOs. Wells containing only modified medium and inoculum served as control groups. The plates were incubated at 37 °C for 24 h, after which the crystal violet binding assay was performed to assess biofilm formation. The percentage reduction in biofilm was calculated using the following formula:Reduction of biofilm formation (%) = [(C − B) − (T − B)/(C − B)] × 100
where C represents the absorbance value of the well containing only the bacterial culture and medium, B the absorbance value of the well containing the modified medium, and T the absorbance value of the well containing both the culture and the EOs at different concentrations. Absorbance measurements were taken at 595 nm using a microplate reader (BioTek Elisa reader, μQuant, BioTek Inc., Winooski, VT, USA).

### 3.6. Anti-Quorum Sensing Activity Test

The anti-QS activity was assessed using the reporter strain *Chromobacterium violaceum* ATCC 12472 to evaluate the inhibitory effects of the essential oils. Prior to the assay, the MIC value of each essential oil was determined. The test was performed following a modified version of the method outlined by Batohi et al. (2021) [41]. *C. violaceum* ATCC 12472 was cultured in Luria-Bertani (LB) broth (Merck, Darmstadt, Germany) supplemented with 0.5% Tween 80 as the control, and in LB broth containing sub-MIC concentrations of the EOs. The cultures were incubated at 30 °C for 24 h. Following incubation, 1 mL of each culture suspension was collected and centrifuged at 10,000× *g* for 5 min. The supernatant was discarded, and the resulting pellets were resuspended in 1 mL of dimethyl sulfoxide (DMSO). Violacein production was quantified by measuring absorbance at 585 nm. The percentage inhibition of violacein production was calculated using the following formula:Violacein inhibition (%) = [(OD585 nmControl − OD585 nmTest)/OD585 nmControl)] × 100 

### 3.7. Statistical Analysis

Data were represented as means ± SD. Statistical differences were analyzed using One-Way ANOVA and Tukey’s test for post-hoc comparisons with GraphPad Prism software (version 8.0, GraphPad, Boston, MA, USA).

## 4. Conclusions

Results revealed that EOs from *T. spicata* subsp. *spicata* are dominantly rich with carvacrol, in agreement with the literature. Both samples showed remarkable antimicrobial, antibiofilm and anti-QS activities, largely due to their high carvacrol content. These results emphasize the potential of *T. spicata* subsp. *spicata* EOs as natural antimicrobial agents, especially in combating biofilm-associated infections and inhibiting QS in pathogenic bacteria. In addition, both samples showed moderate cytotoxic properties against cancer cell lines. This study revealed significant medicinal potential of *T. spicata* subsp. *spicata* essential oil.

## Figures and Tables

**Figure 1 antibiotics-14-00181-f001:**
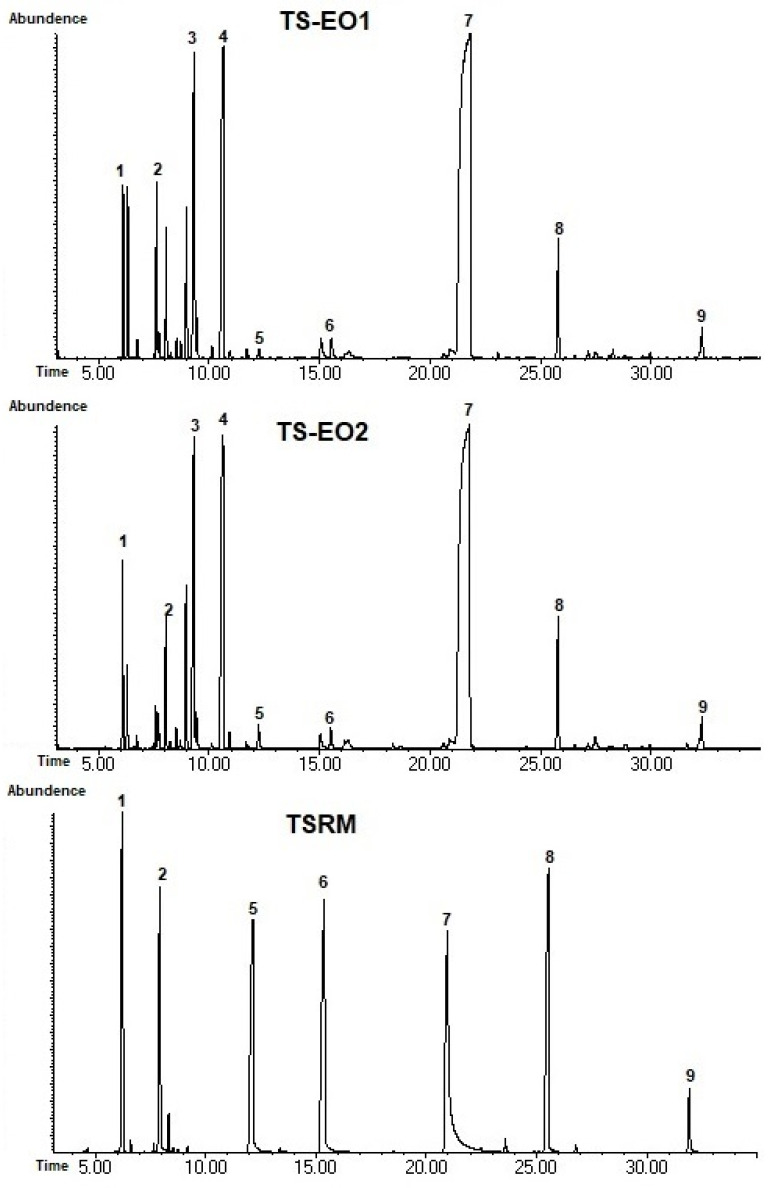
GC-MS chromatograms of the samples and reference mixture. TSRM: *T. spicata* reference mixture. 1: α-pinene, 2: β-myrcene, 3: p-cymene, 4: γ-terpinene, 5: Linalool, 6: terpinene-4-ol, 7: carvacrol, 8: caryophyllene, 9: Caryophyllene oxide.

**Figure 2 antibiotics-14-00181-f002:**
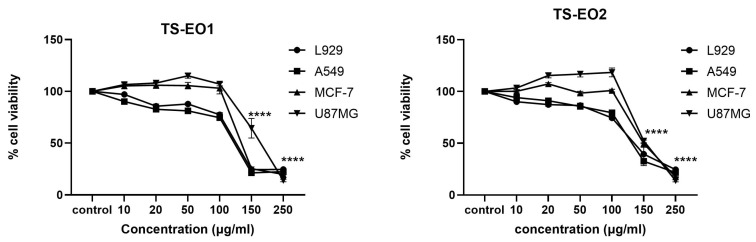
Cytotoxic effect of TS-EO1 and TS-EO2 on A549 (human adenocarcinoma), U87MG 140 (human glioblastoma), MCF-7 (human breast carcinoma) and L929 (mouse fibroblast) cells. Three human tumor cell lines and one normal cell line were exposed with different concentrations. Each bar denotes the mean (±SD) of three independent experiments conducted in triplicate. **** *p* < 0.001.

**Figure 3 antibiotics-14-00181-f003:**
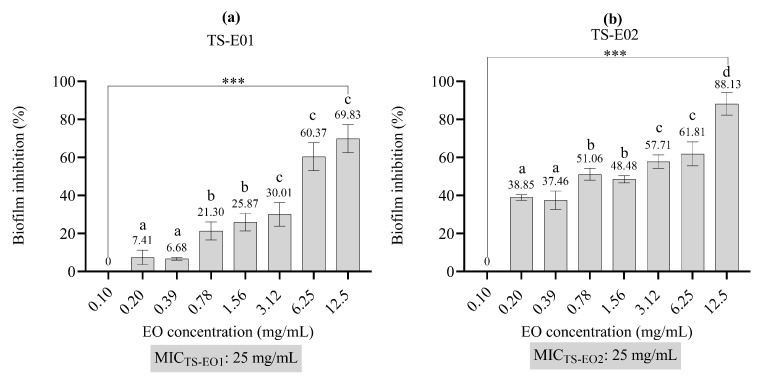
Antibiofilm activity of essential oils. (**a**) TS-EO1. (**b**) TS-EO2. Bars represent standard deviations. One-Way ANOVA and Tukey’s Test were performed to compare the mean values. Different letters indicate the statistical difference between the groups. ***; *p* < 0.001, TS-EO1: *T. spicata* essential oil 1; TS-EO2: *T. spicata* essential oil 2.

**Figure 4 antibiotics-14-00181-f004:**
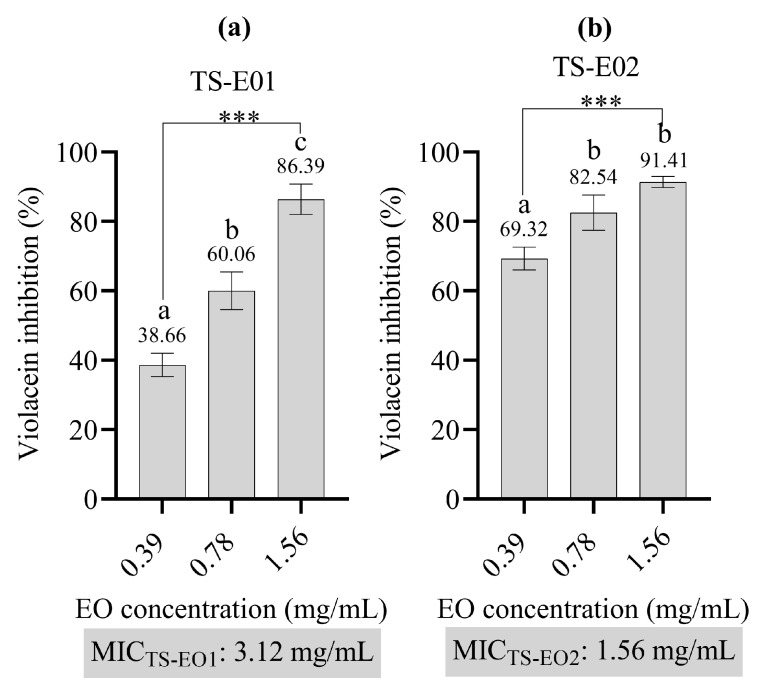
Violacein inhibition (%). (**a**) TS-EO1. (**b**) TS-EO2. Bars represent standard deviations. One-Way ANOVA and Tukey’s Test were performed to compare the mean values. Different letters indicate the statistical difference between the groups. ***; *p* < 0.001, TS-EO1: *T. spicata* essential oil 1; TS-EO2: *T. spicata* essential oil 2.

**Table 1 antibiotics-14-00181-t001:** Essential oil composition of aerial parts of *T. spicata*.

RT ^1^	RRI ^2^	RRI ^3^	Compounds	TS-EO1 (%)	TS-EO2 (%)	IM ^4^
6.114	926	931	α-Thujene	2.7	3.0	a, b
6.319	933	939	α-Pinene	2.8	1.3	a, b, c
6.753	948	953	Camphene	0.3	0.2	a, b
7.625	977	980	β-pinene	3.3	0.8	a, b
7.738	981	979	1-Octen-3-ol	0.6	0.7	a, b
8.062	992	991	β-Myrcene	2.3	2.4	a, b, c
8.534	1006	1005	α-Phellandrene	0.4	0.4	a, b
8.724	1011	1011	δ-3-carene	0.3	-	a, b
8.974	1018	1018	α-Terpinene	3.1	3.4	a, b
9.329	1027	1026	p-Cymene	10.1	12.3	a, b
9.442	1030	1031	β-Phellandrene	0.9	-	a, b
10.132	1048	1040	β-cis-Ocimene	0.3	-	a, b
10.63	1062	1062	γ-Terpinene	12.4	13.3	a, b
10.928	1069	1068	Trans-Sabinene hydrate	-	0.4	a, b
12.261	1104	1098	Linalool	0.3	0.7	a, b, c
15.067	1171	1169	endo-Borneol	0.9	0.7	a, b
15.532	1182	1177	Terpinen-4-ol	0.8	0.7	a, b, c
21.795	1328	1298	Carvacrol	54.3	54.1	a, b, c
25.791	1423	1418	Caryophyllene	3.1	3.6	a, b, c
27.456	1464	1493	p-tert-Butylcatechol	-	0.5	a, b
32.157	1584	1576	Spathulenol	0.1	0.1	a, b
32.298	1588	1581	Caryophyllene oxide	1.0	1.1	a, b, c
53.208	2224	2253	Pimara-7,15-dien-3-ol	-	0.3	a, b
			Total identified	100.0	100.0	

^1^ RT: Retention time. ^2^ RRI: Linear program retention indices determined on HP-5MS (30 m) column. ^3^ RRI: RRI results from the literature. ^4^ IM: Identification method: a: Library, b: RI values from literature, c: Reference compounds.

**Table 2 antibiotics-14-00181-t002:** Anticancer activity of TS-EO1 and TS-EO2 (IC50, μg/mL).

Cell Lines				
	A549 (Human Adenocarcinoma)	U87MG (Human Glioblastoma)	L929 (Mouse Fibroblast)	MCF-7 (Human Breast Carcinoma)
TSEO1	116.3	170.7	127.3	145.5
TSEO2	134.4	150.3	138.6	150.3

**Table 3 antibiotics-14-00181-t003:** Minimum inhibitory concentration (MIC) values of the essential oils of *T. spicata* against tested bacteria (mg/mL).

	Gram-Positive Bacteria			Gram-Negative Bacteria	
	*S. aureus* ATCC 29213 (MSSA)	*S. aureus* ATCC 43300 (MRSA)	*E. faecalis* ATCC 29212	*E. coli* ATCC 25922	*P. aeruginosa* ATCC 27853
TS-EO1	12.5	6.25	25	25	50
TS-EO2	12.5	6.25	12.5	25	50
DMSO (10%)	–	–	–	–	–
Ciprofloxacin	<0.00025	0.0005	0.0625	<0.00025	<0.00025

TS-EO1: *T. spicata* essential oil 1; TS-EO2: *T. spicata* essential oil 2; DMSO: Dimethyl sulfoxide; ATCC: American Type Culture Collection; MSSA: methicillin-susceptible *Staphylococcus aureus*; MRSA: methicillin-resistant *S. aureus*; –: represents no activity.

**Table 4 antibiotics-14-00181-t004:** Minimum inhibitory concentration (MIC) values of the essential oils of *T. spicata* against the tested fungi (mg/mL).

	Fungi				
	*C. parapsilosis* RSKK 994	*C. parapsilosis* ATCC 22019	*C. glabrata* RSKK 4019	*C. krusei* RSKK 3016	*C. albicans* ATCC 10231
TS-EO1	0.20	3.12	50	3.12	6.25
TS-EO2	0.39	1.56	50	0.78	3.12
DMSO (10%)	–	–	–	–	–
Amphotericin B	0.0005	0.00025	0.0005	0.001	0.00025

TS-EO1: *T. spicata* essential oil 1; TS-EO2: *T. spicata* essential oil 2; DMSO: Dimethyl sulfoxide; ATCC: American Type Cuture Collection; RSKK: Refik Saydam National Type Culture Collection; –: represents no activity.

## Data Availability

The data that support the findings of this study are available from the corresponding author upon reasonable request.

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
