# Peer review of "Antimicrobial, Anti-Biofilm, Anti-Quorum Sensing and Cytotoxic Activities of Thymbra spicata L. subsp. spicata Essential Oils"

_antibiotics, 2025, doi:10.3390/antibiotics14020181_

Round 1

Reviewer 1 Report

Comments and Suggestions for Authors

This study evaluated the antimicrobial, anti-biofilm, anti-quorum sensing, and cytotoxic activities of Thymbra spicata subsp. spicata essential oils (TS-EO1 and TS-EO2) from two Turkish localities. TS-EOs exhibited notable antimicrobial activity, especially against Staphylococcus aureus and Candida parapsilosis, with significant biofilm and quorum sensing inhibition. Cytotoxicity was moderate, with IC50 values of 116.3–134.4 μg/mL for A549 cells. These findings highlight the medicinal potential of T. spicata essential oils, marking the first report of their anti-biofilm and anti-quorum sensing properties. The authors assert novelty in unveiling the anti-biofilm and anti-quorum sensing properties of Thymbra spicata essential oils. While the manuscript is well-structured and supported by rigorously designed experiments, several areas require improvement to enhance its clarity and impact:

1.     Keywords: Revise to avoid redundancy with the title.

2.     Botanical Nomenclature: Ensure the botanical name is italicized throughout the manuscript (e.g., page 2, lines 56, 58, 66).

3.     Introduction: Incorporate a clear research gap and objectives, specifically addressing the need for antimicrobial, anti-biofilm, and anti-quorum sensing oils.

4.     Cytotoxicity Comparison: The anticancer activity of the oils should be compared against control cells to assess potential toxicity to healthy tissues.

5.     Figure Legend: Include details of the L929 cell line in the legend of Figure 2.

6.     Oil Concentration: Specify the concentrations of the essential oils used in the experiments.

7.     MIC Justification: The high MIC values (up to 50 mg/mL) compared to standard antibiotics and antifungals should be justified.

             8.   Discussion: Reorganize to better highlight the study’s novelty and clinical relevance

Author Response

Responses to Reviewer 1 Comments

Thank you very much for taking the time to review this manuscript. Please find the detailed responses below and the corresponding revisions in track changes in the re-submitted files.

Comment 1: Keywords: Revise to avoid redundancy with the title.

Response 1: We have revised the keywords to avoid redundancy with the title while ensuring they reflect the core aspects of the study. The revised keywords are as follows: Thymbra spicata subsp. spicata, quorum sensing inhibition, biofilm inhibition, cytotoxicity, essential oil. These changes can be found in the "Keywords" section of the manuscript.

Comment 2: Botanical Nomenclature: Ensure the botanical name is italicized throughout the manuscript (e.g., page 2, lines 56, 58, 66).

Response 2: We have carefully reviewed the manuscript to ensure that all instances of botanical names are italicized as per scientific convention. Specifically, the botanical names on page 2 (lines 56, 58, and 66) have been corrected. Additionally, the manuscript has been thoroughly checked to ensure consistency in italicization throughout.

Comment 3: Introduction: Incorporate a clear research gap and objectives, specifically addressing the need for antimicrobial, anti-biofilm, and anti-quorum sensing oils.

Response 3: Thank you for this valuable suggestion. We have revised the introduction and the following paragraph and references have been added to the manuscript.

“Despite these findings, there is a lack of comprehensive studies investigating the anti-biofilm and anti-quorum sensing properties of T. spicata essential oils, which are critical for combating biofilm-associated infections and antibiotic resistance. Biofilm formation is a significant challenge in clinical settings, as it provides a protective niche for pathogens, enhancing their survival and resistance to conventional antibiotics [10]. Similarly, quorum sensing, a bacterial communication system that regulates biofilm formation and virulence, represents an attractive target for novel antimicrobial strategies [11].

New References Added:

  1. Costerton, J.W.; Stewart, P.S.; Greenberg, E.P. Bacterial biofilms: A common cause of persistent infections. Science 1999, 284(5418), 1318-1322. [https://doi.org/10.1126/science.284.5418.1318]
  2. Fuqua C; Parsek, M.R; Greenberg, E.P. Regulation of gene expression by cell-to-cell communication: acyl-homoserine lactone quorum sensing. Annu Rev Genet 2001, 35, 439-68. [https://doi.org/10.1146/annurev.genet.35.102401.090913]”

Comment 4: Cytotoxicity Comparison: The anticancer activity of the oils should be compared against control cells to assess potential toxicity to healthy tissues.

Response 4: The cytotoxicity assay conducted to evaluate the anticancer potential was performed on three distinct cancer cell lines and one normal cell line. The observed moderate cytotoxic effects emphasize the need for further in-depth studies to better understand the underlying mechanisms and therapeutic implications.

Comment 5: Figure Legend: Include details of the L929 cell line in the legend of Figure 2.

Response 5: Thank you for this valuable suggestion. We have updated the figure legend for Figure 2 as follows.

“Figure 2. Cytotoxic effect of TS-EO1 and TS-EO2 on A549 (human adenocarcinoma), U87MG 140 (human glioblastoma), MCF-7 (human breast carcinoma) and L929 (mouse fibroblast) cells. Three human tumor cell lines and one normal cell line were exposed with different concentrations. Each bar denotes the mean (± SD) of three independent experiments conducted in triplicate

Comment 6: Oil Concentration: Specify the concentrations of the essential oils used in the experiments.

Response 6: Thank you for pointing out this important aspect. We have reviewed the manuscript and ensured that the concentrations of the essential oils used in each experiment are explicitly stated. Specifically:

For antimicrobial activity tests, the concentrations ranged from 50 mg/mL to 0.78 mg/mL, as detailed in the "Antimicrobial Activity Test" subsection of the "Materials and Methods" section.

For antibiofilm activity, sub-MIC concentrations (ranging from 0.20 mg/mL to 12.5 mg/mL) were used, and these have been clarified in the "Antibiofilm Activity Test" subsection.

For cytotoxicity assays, essential oils were tested at concentrations ranging from 10 µg/mL to 250 µg/mL, as detailed in the "Evaluation of Cytotoxicity of EOs" subsection.

Comment 7: MIC Justification: The high MIC values (up to 50 mg/mL) compared to standard antibiotics and antifungals should be justified.

Response 7: Thank you for this valuable suggestion. We have revised the “Results and Discussion-2.3. Evaluation of Antimicrobial Activity” and the following paragraph has been added to the manuscript.

“While the MIC values of T. spicata EOs appear high compared to standard antibi-otics and antifungals, this observation is consistent with previous studies on plant-derived natural products. EOs are complex mixtures of bioactive compounds that function through multi-target mechanisms, often requiring higher concentrations to achieve effective microbial inhibition. Unlike synthetic antibiotics, which are de-signed to act on specific targets, EOs disrupt multiple cellular processes, including membrane integrity, protein synthesis, and QS regulation. Additionally, the hydrophobic nature of essential oils influences their solubility and bioavailability in in vitro conditions, which may contribute to the higher MIC values observed. Despite this, essential oils offer advantages such as lower potential for resistance development and synergistic effects with conventional antimicrobials, making them promising candidates for future therapeutic applications [27]”

Comment 8: Discussion: Reorganize to better highlight the study’s novelty and clinical relevance.

Response 8: We have revised the “Results and Discussion-2.3. Evaluation of Antimicrobial Activity” and the following paragraph has been added to the manuscript.

The findings of this study highlight the significant antimicrobial, anti-biofilm and anti-QS properties of T. spicata EOs, underscoring their potential for application in clinical and pharmaceutical settings. Given the increasing global concern regarding antibiotic-resistant infections, the observed inhibition of QS suggests that T. spicata EOs could serve as alternative agents for reducing bacterial virulence.”

Comment 9: Does the introduction provide sufficient background and include all relevant references?

Response 9: Thank you for your valuable observation. We have reviewed the "Introduction" section to ensure that it provides a comprehensive background and includes all relevant references.

Reviewer 2 Report

Comments and Suggestions for Authors

1. General comment

In the manuscript, the author found that T. spicata essential oils, TS-EO1 and TS-EO2, showed remarkable antimicrobial, antibiofilm and anti-quorum sensing activities, largely due to their high carvacrol content. In addition, both samples showed moderate cytotoxic properties against cancer cell lines. The results suggest the medicinal and pharmaceutical potentials of T. spicata essential oils.

2. Major revision

1) Figure 1: It is recommended to add units for the X and Y axes of the chart.

2Line 271It is recommended to explain the analytical methods of GC-MS and GC-FID, in addition to enter their manufacturer and model names.  

3. Minor revision

1) Line 141: Write the name of tumor cell, mouse fibroblast, in blankets of L929 () cells.

Author Response

Responses to Reviewer 2 Comments

Thank you very much for taking the time to review this manuscript. Please find the detailed responses below and the corresponding revisions in track changes in the re-submitted files.

Comment 1: Figure 1: It is recommended to add units for the X and Y axes of the chart.

Response 1: Thank you for pointing this out. We have revised Figure 1 to include the appropriate units for the X and Y axes.

Specifically:

The X-axis now includes the unit “Time” providing clarity on the time scale of the gas chromatography analysis.

The Y-axis has been labeled as “Abundence” indicating the intensity of the detected compounds.

The updated figure has been incorporated into the revised manuscript.

Comment 2: Line 271:It is recommended to explain the analytical methods of GC-MS and GC-FID, in addition to enter their manufacturer and model names. 

Response 2: We have added the manufacturer and model name "Agilent 5975C Inert XL EI/CI MSD System" in the "3.2. Phytochemical Analysis of EOs" section.

Comment 3: Line 141: Write the name of tumor cell, mouse fibroblast, in blankets of L929 () cells.

Response 3: We have updated the “cell lines used” section in Table 2 to include A549 (human adenocarcinoma), U87MG (human glioblastoma), MCF-7 (human breast carcinoma), and L929 (mouse fibroblast).

Reviewer 3 Report

Comments and Suggestions for Authors

Here are my comments and suggestions:

  1. The author should italicize the names of plants and microbes.

  2. Include fluorescence microscopic data for the analysis of the cytotoxicity of essential oils (EOs).

  3. Provide scanning electron microscopy (SEM) or fluorescence microscopy images to examine the biofilm architecture of the pathogen treated with EOs.

  4. Provide the mass fragmentation patterns of each molecule identified in the GC-MS spectra as a supplementary file.

  5. Discuss the antimicrobial roles of the compounds listed in Table 1.

  6. What is the mechanism of anticancer activity of the essential oils (EOs)?

Author Response

Responses to Reviewer 3 Comments

Thank you very much for taking the time to review this manuscript. Please find the detailed responses below and the corresponding revisions in track changes in the re-submitted files.

Comment 1: The author should italicize the names of plants and microbes.

Response 1: Thank you for bringing this to our attention. We have carefully reviewed the manuscript to ensure that all instances of plant and microbial names are italicized as per scientific conventions.

Comment 2: Include fluorescence microscopic data for the analysis of the cytotoxicity of essential oils (EOs).

Response 2: Thank you for this valuable suggestion. Unfortunately, obtaining fluoresence data is not possible at this stage, as we no longer have the essential oil samples used in this study. Regenerating these essential oils could result in compositional variations, potentially altering the experimental outcomes and affecting the reliability of our findings.

Comment 3: Provide scanning electron microscopy (SEM) or fluorescence microscopy images to examine the biofilm architecture of the pathogen treated with EOs.

Response 3: Thank you for this valuable suggestion. Unfortunately, performing additional SEM or fluorescence microscopy analyses is not feasible at this stage, as we no longer have the essential oil samples used in this study. Regenerating these essential oils could result in compositional variations, potentially altering the experimental outcomes and affecting the reliability of our findings.

Comment 4: Provide the mass fragmentation patterns of each molecule identified in the GC-MS spectra as a supplementary file.

Response 4: We have added the mass spectrum of the compounds as a supplementary file.

Comment 5: Discuss the antimicrobial roles of the compounds listed in Table 1.

Response 5: We have discussed the antimicrobial roles of the major compounds listed in Table 1 in the Results and Discussion section. (lines 216-263). Thank you for this valuable suggestion.

Comment 6: What is the mechanism of anticancer activity of the essential oils (EOs)?

Response: Due to the moderate cytotoxic effects observed in different cancer cell lines, further detailed studies at the molecular level are required to investigate whether it possesses anticancer activity.

Round 2

Reviewer 1 Report

Comments and Suggestions for Authors

Although, the author addresses all comments very well, however, I do not get how the authors calculate the concentration of the oils in mg/mL. Also, what solvent is used in it. Furthermore, the toxicity (If the oil is cytotoxic or not) assay of the oil should assessed up to the MIC concentration.  

Author Response

Responses to Reviewer 1 Comments

Comment: Although, the author addresses all comments very well, however, I do not get how the authors calculate the concentration of the oils in mg/mL. Also, what solvent is used in it. Furthermore, the toxicity (If the oil is cytotoxic or not) assay of the oil should assessed up to the MIC concentration. 

Response: We thank the reviewer for pointing out that we need to clarify how we calculated the mg/mL concentrations of our essential oils.

For antimicrobial testing the concentrations of the essential oils (EOs) were determined by accurately weighing 100 mg of EO using an analytical balance and dissolving it in a known volume of 10% DMSO and 0.5% Tween 80, yielding a 100 mg/mL stock solution. This stock solution was vortexed thoroughly to ensure homogeneity and subsequently diluted to the required concentrations for antimicrobial and antibiofilm assays. For the antimicrobial and antibiofilm assays, the appropriate media supplied with 0.5% Tween 80 was also used. We chose DMSO and Tween 80 because it is commonly used for in vitro studies due to their ability to dissolve essential oils.

Tween 80 is known for its ability to stabilise emulsions due to its high hydrophilic-lipophilic balance (HLB). This property makes it possible to effectively solubilise essential oils in aqueous solutions, which is crucial as essential oils usually have low water solubility and displace lipophilic components.

DMSO is also often used to dissolve essential oils as it can penetrate biological membranes and improve the bioavailability of poorly soluble compounds. It is known that the preparation of essential oil solutions with 10% DMSO facilitates their use in bioassays, demonstrating its effectiveness as a solvent in experimental setups. Studies show that DMSO is generally considered non-toxic at concentrations below 10%. This makes it a popular choice for in vitro studies where the safety of the solvent is critical. Also, when evaluating the antimicrobial effect of DMSO at concentrations below 10, no antimicrobial effect was observed (see Tables 3 and 4).

For cytotoxicity assays, both oils were dissolved in absolute ethanol at a concentration of 2.5 mg/100 µl to prepare a stock solution. From this stock solution, a growth medium containing 250 µg/ml of essential oil with 1% ethanol was prepared. Dilutions were then made to obtain essential oil concentrations of 10, 20, 50, 100, and 150 µg/ml.

We appreciate the reviewer's valuable suggestion regarding cytotoxicity assessment at MIC concentrations. However, we would like to clarify that the MIC values in our study were reported in mg/mL, while the cytotoxicity assays were conducted in µg/mL. This difference in concentration units means that assessing cytotoxicity at MIC levels would require testing at significantly higher doses than typically used in cytotoxicity assays.

References

  1. Elfiyani, R., Amalia, A., & Pratama, S. Y. (2017). Effect of using the combination of tween 80 and ethanol on the forming and physical stability of microemulsion of eucalyptus oil as antibacterial. Journal of Young Pharmacists, 9(1s), s1.
  2. Eesiah, S., Yu, J., Dingha, B., Amoah, B., & Mikiashvili, N. (2022). Preliminary assessment of repellency and toxicity of essential oils against Sitophilus zeamais motschulsky (Coleoptera: Curculionidae) on stored organic corn grains. Foods, 11(18), 2907.
  3. Onder, A., Rızvanoğlu, S. S., Gündoğu, E. F., Demirci, B., & Eryilmaz, M. (2024). Chemical composition and antimicrobial, anti-biofilm, and anti-quorum sensing activities of Mentha longifolia subsp. typhoides essential oil. Plant Biosystems-An International Journal Dealing with all Aspects of Plant Biology, 158(5), 1085-1092.

Reviewer 3 Report

Comments and Suggestions for Authors

well done

Author Response

Response to Reviewer 3: We sincerely appreciate the reviewer's positive feedback. Thank you for taking the time to evaluate our manuscript. Your insights and comments have been valuable in improving the quality of our work.